# Effect of the Functional VP1 Unique Region of Human Parvovirus B19 in Causing Skin Fibrosis of Systemic Sclerosis

**DOI:** 10.3390/ijms242015294

**Published:** 2023-10-18

**Authors:** Der-Yuan Chen, Chih-Chen Tzang, Chuan-Ming Liu, Tsu-Man Chiu, Jhen-Wei Lin, Pei-Hua Chuang, Chia-Wei Kuo, Bor-Show Tzang, Tsai-Ching Hsu

**Affiliations:** 1Institute of Medicine, Chung Shan Medical University, Taichung 402, Taiwan; dychen1957@gmail.com (D.-Y.C.); santafe1062@gmail.com (C.-M.L.); emmachiu@csmu.edu.tw (T.-M.C.); weiwei870710@gmail.com (J.-W.L.); dylan.ymu@gmail.com (P.-H.C.); kgw022096507@gmail.com (C.-W.K.); 2College of Medicine, China Medical University, Taichung 404, Taiwan; 3Translational Medicine Laboratory, Rheumatology and Immunology Center, China Medical University Hospital, Taichung 404, Taiwan; 4School of Medicine, College of Medicine, National Taiwan University, Taipei City 100, Taiwan; jerrytzang@gmail.com; 5Department of Dermatology, Chung Shan Medical University Hospital, Taichung 402, Taiwan; 6School of Medicine, Chung Shan Medical University, Taichung 402, Taiwan; 7Department of Biochemistry, School of Medicine, Chung Shan Medical University, Taichung 402, Taiwan; 8Department of Clinical Laboratory, Chung Shan Medical University Hospital, Taichung 402, Taiwan; 9Immunology Research Center, Chung Shan Medical University, Taichung 402, Taiwan

**Keywords:** human parvovirus B19 (B19), systemic sclerosis (SSc), VP1 unique region (VP1u), macrophages, fibrosis

## Abstract

Human parvovirus B19 (B19V) is a single-stranded non-enveloped DNA virus of the family Parvoviridae that has been associated with various autoimmune disorders. Systemic sclerosis (SSc) is an autoimmune connective tissue disorder with high mortality and has been linked to B19V infection. However, the precise mechanism underlying the B19V contribution to the development of SSc remains uncertain. This study investigated the impacts of the functional B19V-VP1 unique region (VP1u) in macrophages and bleomycin (BLE)-induced SSc mice. Cell experimental data showed that significantly decreased viability and migration of both B19V-VP1u-treated U937 and THP-1 macrophages are detected in the presence of celastrol. Significantly increased MMP9 activity and elevated NF-kB, MMP9, IL-6, TNF-α, and IL-1β expressions were detected in both B19V-VP1u-treated U937 and THP-1 macrophages. Conversely, celastrol revealed an inhibitory effect on these molecules. Notably, celastrol intervened in this pathogenic process by suppressing the sPLA2 activity of B19V-VP1u and subsequently reducing the inflammatory response. Notably, the administration of B19V-VP1u exacerbated BLE-induced skin fibrosis in mice, with augmented expressions of TGF-β, IL-6, IL-17A, IL-18, and TNF-α, ultimately leading to α-SMA and collagen I deposits in the dermal regions of BLE-induced SSc mice. Altogether, this study sheds light on parvovirus B19 VP1u linked to scleroderma and aggravated dermal fibrosis.

## 1. Introduction

Human parvovirus B19 (B19V), classified in the genus Erythrovirus within the Parvoviridae family, features a non-enveloped icosahedral structure with linear single-stranded DNA, measuring 23–26 nm in diameter [1,2]. Research has indicated that B19V infections are globally prevalent among various age groups, affecting 30–60% of adults [3,4]. The B19V genome encodes a single-stranded DNA approximately 5.6 kb in length, giving rise to a capsid mainly consisting of VP1 (83 kDa) and VP2 (58 kDa) proteins. Located at the amino terminus of VP1, the VP1 unique region (VP1u) plays crucial roles in viral tropism, internalization, cellular transport, and nuclear localization [5,6,7]. Indeed, B19V-VP1u has been associated with many disorders such as adult-onset Still’s disease, systemic lupus erythematosus, rheumatoid arthritis, and asthma [8,9,10,11]. Additionally, B19V-VP1u encompasses a phospholipase A2 (PLA2) domain that catalyzes phospholipid hydrolysis, releasing lipid mediators vital for inflammation and host defense, also bolstering its survival by escaping endocytic vesicles during cell entry [1,12].

Systemic sclerosis (SSc), namely systemic scleroderma, is an autoimmune connective tissue disorder marked by excessive collagen deposition in the skin and organs, contributing to high mortality [13]. SSc and the key diagnostic criteria are skin sclerosis and Raynaud’s phenomenon, with subsequent organ metabolic disruptions including vessel abnormalities and widespread fibrosis [14,15]. Within pathological processes of SSc, immune cells release cytokines such as TGF-β, IFN-α, IL-13, TNFα, and IL-1, which play vital roles in inflammation and fibrosis [16]. In addition, macrophages are key sources of platelet-derived growth factor (PDGF) [17], which, along with TGF-β, contributes to excessive collagen production and fibrosis [18]. Moreover, α-SMA and collagen I are also known as markers of myofibroblast production [19]. Interestingly, prior research has indicated an association between B19V and SSc dermal fibrosis [13,20,21,22]. However, the precise mechanism underlying the B19V contribution to the development of SSc remains uncertain.

Derived from traditional Chinese medicinal plants, celastrol is renowned for its anti-inflammatory, antioxidant, and anticancer properties [23]. Evidence has demonstrated the preventive and therapeutic potential of celastrol across a spectrum of chronic inflammatory and autoimmune disorders [24,25,26]. Additionally, celastrol modulates inflammation through the inhibition of secretory phospholipase A2 group IIA [27]. Accordingly, celastrol has revealed significant inhibitory potentials on sclerosis in various organs [28,29,30]. Although a recent finding reported the association between B19V infection and SSc via activation of dermal fibroblasts [22], very limited information about the role of functional B19V-VP1u in the pathogenesis of fibrosis is known. The current study utilized a bleomycin-induced scleroderma fibrosis mouse model to investigate the impacts of functional B19V-VP1u recombinant proteins on scleroderma skin fibrosis. Concurrent administration of the drug celastrol was also employed to understand its effects on B19V-VP1u-secreted phospholipase A2 (sPLA2) activity and bleomycin-induced SSc skin fibrosis.

## 2. Results

### 2.1. Effects of sPLA2 Activity of B19V-VP1u on Human Macrophage

The inhibitory effect of celastrol on secretory phospholipase A2 group IIA is well known [27], but its effect on group XIII sPLA2 of parvovirus B19 VP1u [1] remains unknown. A sPLA2 activity assay kit was adopted to assess the effect of celastrol on the sPLA2 activity of B19V-VP1u according to manufacturers’ instructions. Bee venom PLA2 (bvPLA2) served as the sPLA2 activity positive control. As shown in Table 1, significantly decreased bvPLA2 activity of bee venom was detected in the presence of celastrol in a dose-dependent manner. Similar results were observed, such that significantly decreased sPLA2 activity of B19V-VP1u was detected in the presence of celastrol in a dose-dependent manner. Accordingly, no sPLA2 activity was detected in the presence of different concentrations of celastrol. To evaluate the cytotoxicity of celastrol on human acute monocytic leukemia U937 and THP-1-derived macrophages, the viability of U937 and THP-1 macrophages treated with different concentrations of celastrol was determined using an MTT assay. In the presence of 1 µg/mL B19V-VP1u, no significant difference in cell viability was observed in both U937 and THP-1 macrophages (Figure 1A,B). Significantly decreased viability of both B19V-VP1u (1 µg/mL)-treated U937 and THP-1 macrophages was detected in the presence of celastrol in a dose-dependent manner with an IC50 value of 1.53 and 1.28 µM, respectively (Figure 1A,B). To further evaluate the effects of celastrol on B19V-VP1u-induced macrophage functions, cell migration assays were performed. Significant inhibitory effects of celastrol on cell migration were detected in both U937 and THP-1 macrophages treated with 1 µg/mL B19V-VP1u (Figure 1C,D).

### 2.2. Effects of B19V-VP1u-Activated Inflammatory Responses in Human Macrophage

To evaluate the effects of inflammatory responses in B19V-VP1u-activated macrophages, the activity of matrix metallopeptidase-9 (MMP-9), expressions of MMP-9 and NF-kB proteins, and concentrations of inflammatory cytokines were measured. Significantly increased MMP-9 activity was detected in B19V-VP1u-activated U937 and THP-1 macrophages. Conversely, MMP-9 activity was significantly reduced in both B19V-VP1u-activated U937 and THP-1 macrophages treated with celastrol in a dose-dependent manner (Figure 2A,B). Significantly decreased MMP-9 and NF-kB proteins based on the amount of GAPDH in both B19V-VP1u-activated U937 and THP-1 macrophages were observed in the presence of celastrol (Figure 2C,D). Additionally, significantly decreased IL-6, TNF-α, and IL-1β levels were detected in the medium of B19V-VP1u-activated U937 and THP-1 macrophages treated with celastrol (Figure 3A–F).

### 2.3. Effects of B19V-VP1u on Skin Tissues in Bleomycin-Induced Systemic Sclerosis (BLE-SSc) Mouse Model

Figure 4A represents the schematic diagram of the BLE-SSc fibrosis mouse model. The animal grouping and BLE-SSc mice were generated as described in the schematic diagram and the Materials and Methods section. The skin thickness of the mice was measured using H&E staining and Masson’s Trichrome staining, indicating increased thickness due to bleomycin-induced SSc fibrosis (Figure 4B,C). A significant increase in dermal thickness was detected in the mice of the BLE group as compared to those of the control group (Figure 4C). Notably, the mice of the BLE + VP1u group showed even more pronounced thickening as compared to mice of the BLE group, signifying exacerbated fibrotic response with B19V-VP1u presence (Figure 4C). In celastrol-treated groups, BLE + Cel and BLE + Cel + VP1u, the mice revealed significant dermal thickness reduction as compared to the mice of BLE and BLE + VP1u groups, respectively (Figure 4C). In addition, hydroxyproline levels were assessed in skin tissues of mice from all groups. Elevated hydroxyproline content in the skin of the mice from BLE, BLE + Celastrol, and BLE + VP1u groups was observed as compared to the mice of the control group (Figure 4D). Moreover, the BLE + VP1u group exhibited significantly higher hydroxyproline levels than the mice of BLE group, suggesting aggravated fibrosis due to the B19V-VP1u protein. However, celastrol administration in the mice of BLE + Cel and BLE + VP1u + Cel groups effectively attenuated both bleomycin-induced fibrosis and the aggravated fibrotic impact of the B19V-VP1u protein (Figure 4D).

### 2.4. Effects of B19V-VP1u on α-SMA, Collagen I, and Cytokine Expressions in Skin Tissues of Bleomycin-Induced Systemic Sclerosis (BLE-SSc) Mouse Model

To further assess the fibrotic state of the skin in BLE-treated mice, quantitative analysis was conducted on IHC-stained tissues using a digital image analysis system as described in Materials and Methods. Significantly increased positive cell count of TGF-β, α-SMA, and collagen I were observed in IHC-stained tissues of mice from the BLE and BLE + VP1u groups as compared to those from the control group (Figure 5). A significantly increased positive cell count of IL-6, IL-17A, IL-18, and TNF-α was observed in IHC-stained tissues of mice from the BLE and BLE + VP1u groups as compared to those from the control group (Figure 6). Notably, a significantly higher positive cell account of TGF-β, α-SMA, IL-6, IL-17A, and TNF-α was detected in IHC-stained tissues of mice from the BLE + VP1u groups as compared to those from the BLE group (Figure 5 and Figure 6). Celastrol administration, as shown in mice from the BLE + Cle and BLE + VP1u + Cel groups, consistently led to decreased positive cell count as compared to those of the BLE and BLE + VP1u groups, respectively (Figure 5 and Figure 6). Moreover, fibrosis-related mRNA expressions were also analyzed. Significantly increased mRNA levels of TGF-β, MMP-9, and TNF-α were detected in the skin tissues of mice from the BLE group as compared to those from the control group. Significantly higher TGF-β, MMP-9, and TNF-α mRNA levels were detected in the skin tissues of mice from BLE + VP1u group as compared to those from the BLE group (Figure 7). In celastrol-treated groups, BLE + Cel and BLE + VP1u + Cel, the mice revealed significant reductions in all three inflammatory mRNA levels as compared to those from the BLE and BLE + VP1u groups, respectively (Figure 7).

## 3. Discussion

Studies have consistently demonstrated B19V infection within the bone marrow and skin samples of systemic sclerosis (SSc) patients, potentially acting as a source for the spread of the virus to SSc tissues [2,31,32,33]. However, the precise role of B19V, especially functional B19V-VP1u, in the pathogenesis of SSc is still unclear. Notably, endothelial and fibroblast cells are primary targets for B19V infection, and this persistent infection could contribute to fibroblast aging and fibrosis development [34,35]. Intriguingly, B19V activates normal human dermal fibroblasts (NHDF), inducing gene expressions associated with fibrosis and inflammation, like α-SMA, TGF-β, and IL-1β [22]. Furthermore, B19V’s interaction with the host’s innate immune responses might lead to damage in epithelial and fibroblast cells, leading to fibrosis [36]. In our study, immunohistochemistry (IHC) revealed heightened expressions of fibrotic markers in dermal fibroblast cells following B19V-VP1u exposure, providing rational evidence for explanting the role of B19V, particularly functional B19V-VP1u, in the pathogenesis of SSc.

Moreover, research into viral interactions in SSc revealed the role of B19V infection in generating autoantibodies against multiple autoantigens, suggesting its contribution to the formation of pathologically significant autoantibodies in SSc [20]. For instance, anti-PDGF receptor antibodies present in SSc can induce reactive oxygen species, myofibroblast differentiation, and type I collagen production [37]. A distinct study utilizing bleomycin-induced systemic sclerosis mouse models identified autoantibodies directed at various tissues including skin, lungs, heart, stomach, spleen, and thymus [36]. Though we did not investigate whether B19V-VP1u triggered the development of autoantibodies nor whether celastrol impacted the generation of autoantibodies, it has been demonstrated that celastrol has the capacity to eradicate the downstream inflammation and fibrotic response generated by autoimmunity. We are optimistic that our findings regarding the antifibrotic effects of celastrol could be applicable to various organ systems, given the similar pathogenic responses to fibrosis in these systems, often involving NF-kB as a central factor and comparable downstream reactions [19].

Our research validates that B19V-VP1u is responsible for exacerbating SSc, and its inhibition by celastrol has been demonstrated. VP1u assumes a pivotal role as a significant infectious structure [1], rendering it an ideal subject for exploring the mechanisms behind fibrosis in SSc. Notably, VP1u features a unique PLA2 domain, or group XIII phospholipase A2, associated with viral activity and infection [38,39]. This enzyme catalyzes phospholipid breakdown to generate arachidonic acid, which is a precursor of eicosanoids, prostaglandins, and leukotrienes associated with inflammatory responses, cellular migration, and tissue damage [40,41]. Studies have indicated that persistent infections by pathogens like B19V, HCMV, and HHV-6 could contribute to inflammation and the development of SSc [42]. Adopting an antiviral approach, as demonstrated in our research, could potentially offer a solution to address the underlying root causes of pathogenic SSc. This approach may alleviate the need for conventional immune-suppressing agents, relieving patients from experiencing adverse effects.

## 4. Materials and Methods

### 4.1. sPLA2 Activity of B19V-VP1u

Recombinant B19V-VP1u proteins were prepared as described elsewhere [43], and their sPLA2 activity was measured using a colorimetric assay kit (sPLA2 Activity Kit; Cayman Chemical) under the manufacturer’s instructions. The results are expressed as micromoles per minute per milliliter.

### 4.2. Cell Culture

Two human acute monocytic leukemia cell lines, U937 (BCRC 60435) and THP-1 (BCRC 60430), were purchased from the Bioresource Collection and Research Center (BCRC, Food Industry Research and Development Institute, Hsinchu, Taiwan, ROC). The cell lines were tested for authenticity using short tandem repeat (STR) profiling at the National Cheng Kung University (NCKU) Center for Genomic Medicine. The cells were maintained at 37 °C in a humidified atmosphere of 95% air and 5% CO_2_ in RPMI 1640 medium (Thermo Fisher Scientific, Waltham, MA, USA) supplemented with 10% (*v*/*v*) FBS and 100 units/mL penicillin. To differentiate monocytes into adherent macrophages, the cells were seeded at a density of 1 × 10^5^ cells/well in 24-well plates and incubated at 37 °C for 2 days in the presence of 100 nM phorbol 12-myristate 13-acetate (MilliporeSigma, Burlington, MA, USA). The cells were then washed and incubated in a normal growth medium for another 24 h.

### 4.3. Viability of Cells

The 3(4,5cimethylthiazol2yl)2,5diphenyl tetrazolium bromide (MTT) assay was used to determine cell survival. In each well of a 24-well plate, 2 × 10^5^ cells were cultured overnight at 37 °C. The culture medium was removed after treatment with different concentrations of B19V-VP1u alone or with celastrol (ChemFaces, Wuhan, Hubei, China). Subsequently, MTT reagent (0.5 mg/mL) was added and incubated for another 4 h. To dissolve the formazan crystals, 300 µL dimethyl sulfoxide (DMSO) was added to each well of the plate, and the medium’s absorbance was measured at 570 nm using a microplate reader (SpectraMax M5, Molecular Devices LLC, San Jose, CA, USA).

### 4.4. Cell Migration Assay

To investigate the influence of B19V-VP1u on cell motility, Millicell Hanging Cell Culture inserts (8-µm pore size; EMD Millipore, Burlington, MA, USA) were adopted. In brief, the upper chamber containing the serum-free RPMI medium (5 × 10^5^ cells/well) or different concentrations (0, 0.5, 1, 2 µM) of celastrol was merged with the bottom chamber containing the standard medium (RPMI1640 with 10% FBS). Assembled chambers were then incubated for 24 h at 37 °C. The migrating cells were fixed with 10% neutral buffered formalin at 25 °C for 2 h before being stained with 0.05% Giemsa solution. A light microscope (ZEISS AXioskop 2, Göppingen, GER) at a magnification of ×200 per filter was used to count the number of migrated cells in ten random fields from each experiment.

### 4.5. Zymography Assay

U937 and THP-1 macrophages were incubated with recombinant B19V-VP1u proteins (1 µg/mL), and gelatin-zymography assay was used to measure MMP-9 activity in the medium. An 8% sodium dodecyl sulfate-polyacrylamide gel electrophoresis (SDS-PAGE) gel containing 0.1% gelatin was adopted to separate ten microliters of culture medium from each treatment. After 1 h soaking in 2.5% Triton X-100 to remove the SDS, the gels were rinsed in the reaction buffer, which contained 40 mM Tris-HCl, pH 8.0, 10 mM CaCl2, and 0.02% NaN3. Next, the gels were de-stained with a methanol-acetic acid solution, and the gelatinolytic activity was then visualized after staining with 0.5% Coomassie brilliant blue R-250. A gel documentation and analysis system (Alphaimager HP, Proteinsimple, Bath, UK) was used to measure the relative MMP levels.

### 4.6. Zymography Assay

The concentrations of human interleukin (IL)-1β, IL-6, and tumor necrosis factor (TNF)-α in cell culture media were measured using ELISA kits according to the manufacturer’s instructions (Invitrogen, Thermo Fisher Scientific, Waltham, MA, USA).

### 4.7. Immunoblotting

Protein samples were separated using a 10% SDS-PAGE and subsequently transferred to a PVDF membrane (ImmobilonE, 0.45 µM; MilliporeSigma, Burlington, MA, USA). The membrane was incubated for 3 h with antibodies against MMP-9 (Merk Millipore, Sigma Aldrich, Temecula, CA, USA), NF-kB (ABclonal, Boston, MA, USA), and GAPDH (Novus Biologicals, Centennial, CO, USA), after blocking in 5% non-fat milk for 3 hr. Horseradish peroxidase (HRP)-labeled secondary antibodies were then added and reacted for another hour. Eventually, the immune complexes were visualized and quantified using an Immobilon Western HRP Chemiluminescent Substrate kit (EMD Millipore, Burlington, MA, USA) and densitometry equipment (GE ImageQuant TL 8.1; GE Healthcare Life Sciences, Piscataway, NJ, USA).

### 4.8. Animal Model and Treatments

Fifteen female Balb/c mice at six weeks old were obtained from the National Laboratory Animal Center (NLAC) and housed at the Animal Experimental Center of Chung Shan Medical University. All animal experiments were approved by the Institutional Animal Care and Use Committee of Chung Shan Medical University, Taiwan (approval number: 2584). The bleomycin-induced systemic sclerosis (BLE-SSc) mouse model was generated as described elsewhere [28,44]. The study also incorporated the investigation of the recombinant protein B19V-VP1u [45] and celastrol [46] in the context of mouse fibrosis research. The experiments began when the mice reached eight weeks of age. Firstly, the mice’s back fur was removed, and the animals were randomly divided into five groups, with three mice in each group. Mice in each group were treated as described below: 1. Control group: receiving daily subcutaneous injections of 100 µL saline on the back for five weeks; 2. Bleomycin group (BLE): receiving daily subcutaneous injections of bleomycin (100 μg/100 μL in PBS) on the back starting from day 7 for three weeks; 3. Bleomycin + celastrol group (BLE + Cel): receiving bleomycin as described in group 2 and receiving celastrol (1 mg/kg/day) through intraperitoneal injections for 14 consecutive days starting from day 15; 4. Bleomycin + VP1u group (BLE + VP1u): receiving bleomycin injections as described in group 2 and receiving subcutaneous injections of B19V-VP1u recombinant protein (20 μg/100 μL in PBS) on the back at days 0, 7, 14, and 21; 5. Bleomycin + VP1u + celastrol group (BLE + VP1u + Cel): receiving bleomycin, VP1u, and celastrol as described in groups 3 and 4. All mice were sacrificed at day 28 with CO_2_ euthanasia.

### 4.9. Hydroxyproline Colorimetric Assay

The contents of skin collagen were detected using a Hydroxyproline Assay Kit (Colori metric) (Ab222941, Abcam, Cambridge, UK). Briefly, tissue samples were weighted and pooled with 100 μL ddH_2_O per 10 mg tissue in a 2 mL Eppendorf tube. The tissue was then homogenized using a Polytron RT 3000 tissue homogenizer. Following the protocol, further steps were performed, and the absorbance at 560 nm was measured using a SpectraMax M5 Microplate Reader (SpectraMax M5, Molecular Devices LLC, San Jose, CA, USA).

### 4.10. H&E (Hematoxylin and Eosin) Stain and Masson’s Trichrome Stain

For the H&E stain, the tissue sections were immersed in Hematoxylin stain for 3 min, followed by rinsing in distilled water three times and 1 min immersion in 85% ethanol. Sequentially, the sections were immersed in 70%, 80%, and 90% ethanol, then twice in absolute ethanol for 2 min each, and twice in xylene for 30 s each before air drying. Masson’s trichrome stain was performed according to the protocol of Bancroft [47].

### 4.11. Immunohistochemistry (IHC)

The tissue sections were fixed with acetone for 5 min and air-dried for 10 min. The sections were then immersed in 0.3% H_2_O_2_/PBS for 5 min to remove peroxidase activity. After treating with blocking solution for 1 h, primary antibodies were applied and incubated for 1 h, followed by secondary antibodies (HRP-conjugated goat anti-rabbit/mouse IgG), reacting for another hour. The sections were then stained with DAB for 5 min and rinsed with distilled water for another 10 min. The primary antibodies used in this experiment were TGF-β (sc-146, Santa Cruz), α-SMA (A2547, Sigma-Aldrich), Collagen I (A5786, ABclonal), IL-6 (sc-1265, Santa Cruz), IL-17A (A12454, ABclonal), and TNF-α (A11534, ABclonal).

### 4.12. Tissue Gnostic

The TissueFAX Plus system (TISSUE GNOSTICS, Vienna, Austria) with the HistoQuest software 2.0 was employed for tissue staining analysis. Five 100 μm fields of view were examined per skin section. Dermal thickness was measured in five different regions of H&E-stained and Masson’s trichrome-stained skin slides to assess fibrosis. IHC staining quantified positively stained cells per square millimeter for TGF-β, α-SMA, collagen I, IL-6, IL-17A, IL-18, and TNF-α, using 100 μm views. Statistical comparisons were performed against a saline negative control group.

### 4.13. qPCR Analysis

To assess skin tissue’s mRNA expression of fibrosis-related genes, the StepOnePlus real-time quantitative PCR method was used. About 100 mg of −20 °C stored skin tissue was utilized per sample. Total RNA was extracted using a Trizol reagent, and its concentration and purity were checked with a spectrophotometer. Then, 2 μg of total RNA underwent cDNA synthesis following the kit instructions. The following primers were used for amplifying mouse genes [48]: TGF-β 5′-GCCTGAGTGGCTGTCTTTTGA-3′ (sense), 5′-CACAAGAGCAGTGAGCGCTGAA-3′ (antisense); MMP-9 5′-CTGTCCAGAC CAAGGGTACAGCCT-3′ (sense), 5′-GAGGTATAGTGGGACACATAGTGG-3′ (antisense); TNFα 5′-ATGAGCACAGAAAGCATGATC-3′ (sense), 5′-TACAGGCTTGTCACTCGAA TT-3′ (antisense).

### 4.14. Statistical Analysis

Experimental results are presented as mean ± standard deviation (mean ± SD), statistically analyzed using one-way ANOVA, and graphed using GraphPad Prism 5.0. A significant level of *p* < 0.05 is considered statistically significant.

## 5. Conclusions

The present study firstly demonstrated that celastrol effectively reduces Group XIII sPLA2 activity of B19V-VP1u and suppresses NF-κB and MMP-9 activity in B19V-VP1u-activated human macrophages. In the bleomycin-induced SSc animal model, the administration of B19V-VP1u led to skin thickening accompanied by increased hydroxyproline levels, all of which were mitigated by celastrol treatment. Higher degrees of fibrosis correlated with increased TGF-β expression and elevated levels of other related fibrotic factors, including α-SMA, collagen I, IL-6, IL-17A, IL-18, TNF-α, and MMP-9. In conclusion, this study reveals that B19V-VP1u exacerbates skin fibrosis in SSc, a phenomenon effectively ameliorated by celastrol treatment. In clinical practice, our results suggest the screening of parvovirus B19 infection would be helpful to identify SSc patients at risk of exacerbating skin fibrosis. In addition, the present study provides the evidence supporting celastrol as a potential antifibrotic agent in SSc patients with novel mechanisms of action. In the future, the development of an anti-B19V-VP1u vaccine not only benefits the prevention of parvovirus B19 but also modulates the pathogenic fibrosis in patients with SSc [20,22,35].

## Figures and Tables

**Figure 1 ijms-24-15294-f001:**
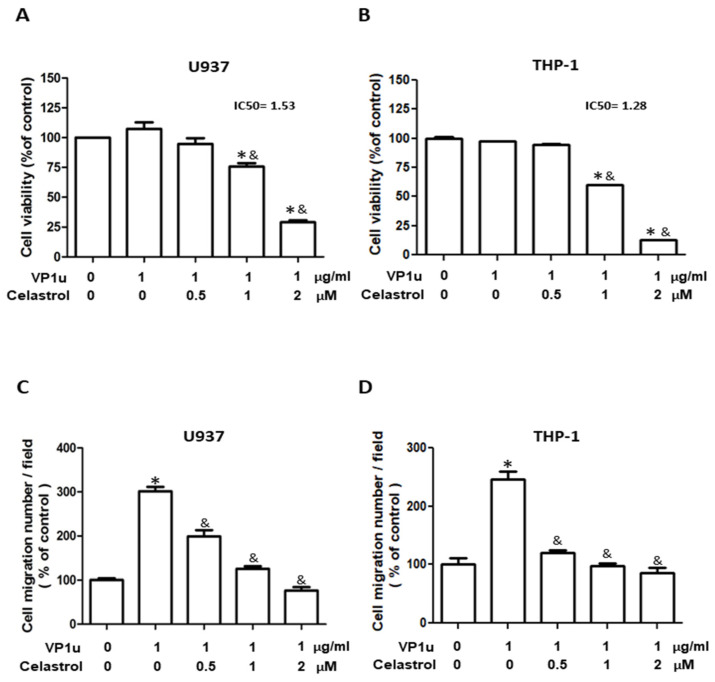
Effects of B19V-VP1u on macrophage survival and migration. Cell viability of (**A**) U937 and (**B**) THP-1 macrophages in the presence of 1 μg/mL B19V-VP1u and different concentrations of celastrol. Cell migration percentage of (**C**) U937 and (**D**) THP-1 macrophages in the presence of 1 μg/mL B19V-VP1u and different concentrations of celastrol. Similar results were obtained in three repeated experiments. The symbols * and & indicate significant differences between the unstimulated control group (0 μM) and the cells treated with 1ug/mL B19V-VP1u alone, respectively.

**Figure 2 ijms-24-15294-f002:**
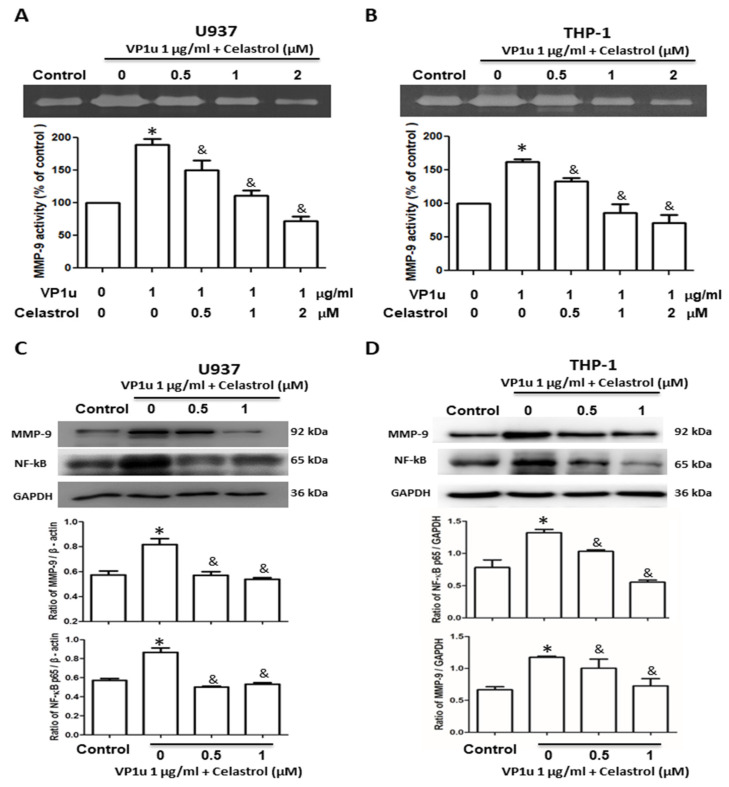
Effects of B19V-VP1u on MMP-9 activity and NF-kB expression in macrophage. MMP-9 activity of (**A**) U937 and (**B**) THP-1 cells in the presence of 1 μg/mL B19V-VP1u and different concentrations of celastrol. Expressions of MMP-9 and NF-kB in (**C**) U937 and (**D**) THP-1 cells in the presence of 1 μg/mL B19V-VP1u and different concentrations of celastrol. Ratios of MMP-9 and NF-kB on GAPDH were shown in the lower panel. Similar results were obtained in three repeated experiments. The symbols * and & indicate significant differences between the unstimulated control group (0 μM) and the cells treated with 1 μg/mL B19V VP1u alone, respectively.

**Figure 3 ijms-24-15294-f003:**
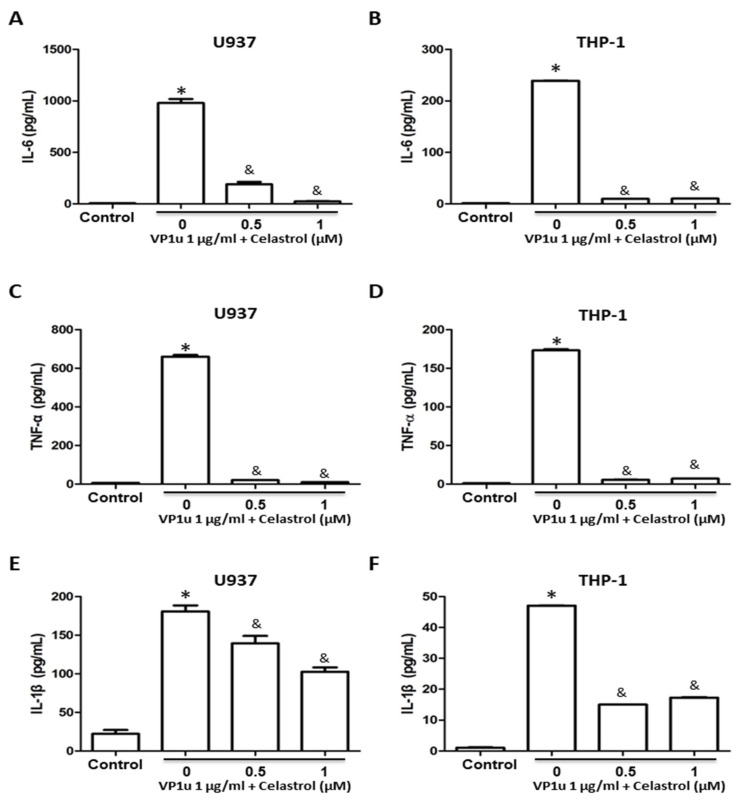
Effects of inflammatory cytokine expressions in B19V-VP1u-activated macrophages. The concentrations of (**A**,**D**) IL-6, (**B**,**E**) TNF-α, and (**C**,**F**) IL-1β in the medium of U937 and THP-1 cells treated with 1 μg/mL B19V-VP1u and different concentrations of celastrol. Similar results were obtained in three repeated experiments. The symbols * and & indicate significant differences between the unstimulated control group (0 μM) and the cells treated with 1 μg/mL B19V VP1u alone, respectively.

**Figure 4 ijms-24-15294-f004:**
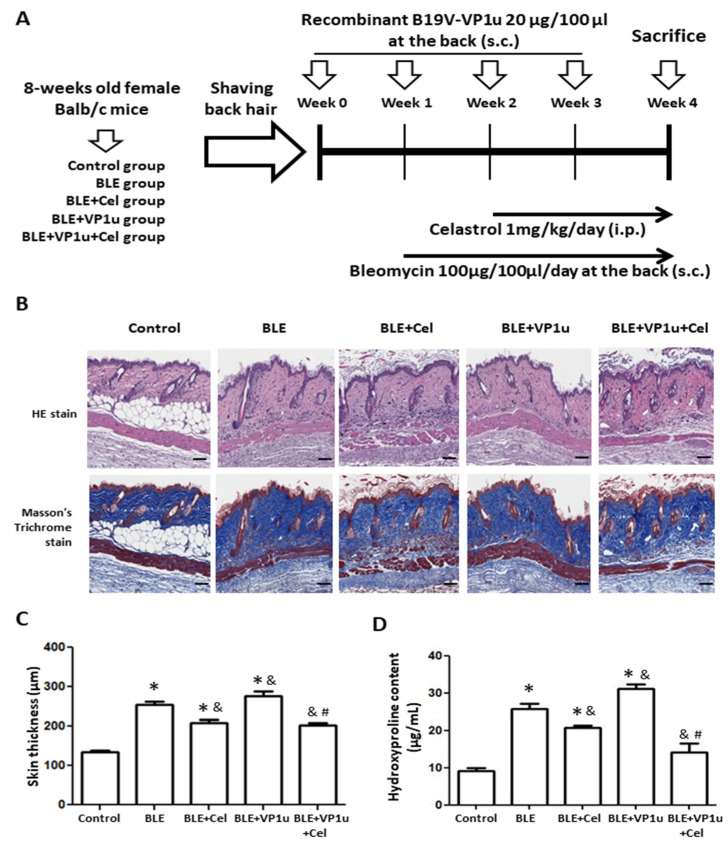
Effects of B19V-VP1u on a mouse model of bleomycin-induced systemic sclerosis (BLE-SSc). (**A**) Schematic diagram of BLE-SSc fibrosis mouse model. (**B**) H&E staining and Masson’s trichrome staining of a mice dermis section from different groups. (**C**) Quantitative representation. (**D**) Hydroxyproline contents of mice dermal thickness from different groups (*n* = 3). Scale bar: 100 μm. The symbols *, &, and # indicate significant differences compared to the control group, BLE group, and BLE + VP1u group, respectively.

**Figure 5 ijms-24-15294-f005:**
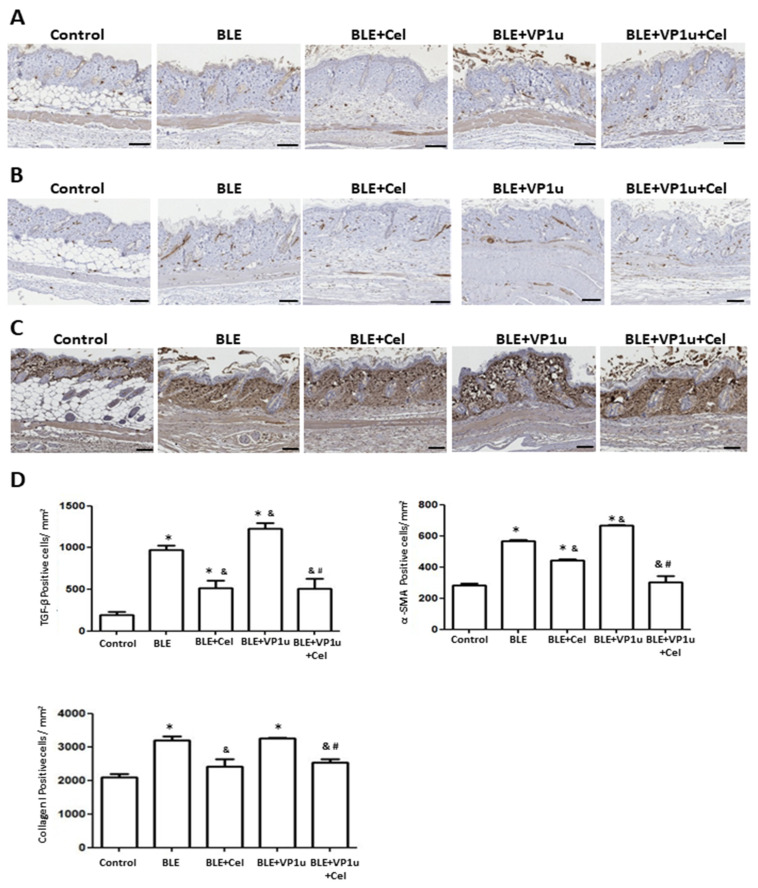
Detection of TGF-β, α-SMA, and collagen I in the back skin of mice. Expressions of (**A**) TGF-β, (**B**) α-SMA, and (**C**) collagen in the back skin of mice from different groups. Scale bar: 100 μm. (**D**) Quantitative results of positive signals. The symbols *, &, and # indicate significant differences compared to the control group, BLE group, and BLE + VP1u group, respectively.

**Figure 6 ijms-24-15294-f006:**
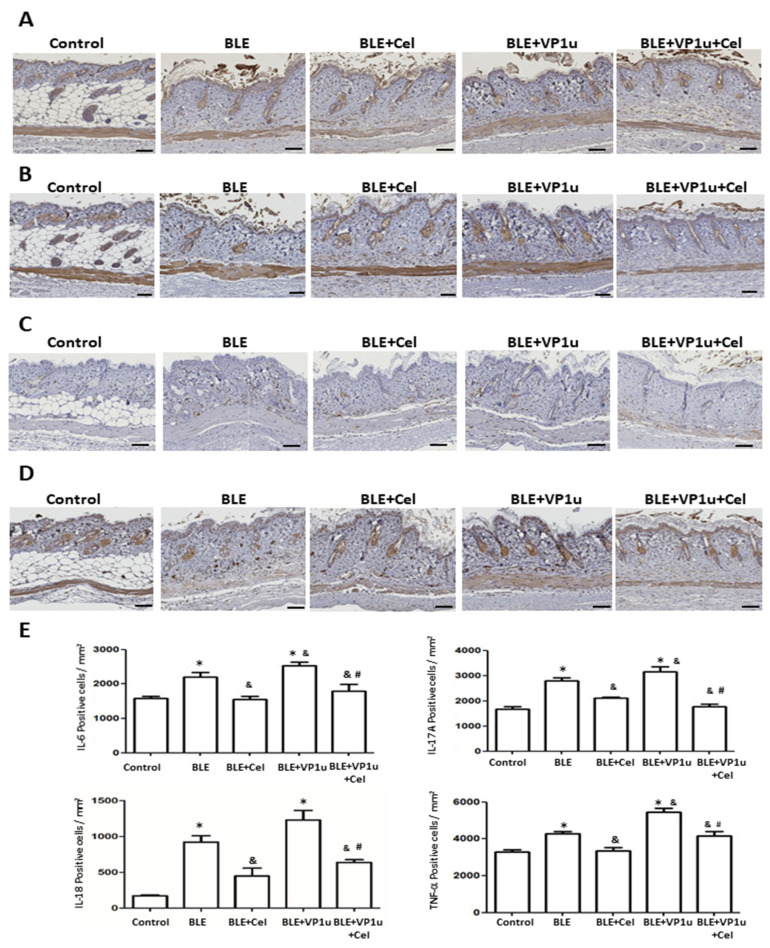
Detection of IL-6, IL-17A, IL-18, and TNF-α in the back skin of mice. Expressions of (**A**) IL-6, (**B**) IL-17A, (**C**) IL-18, and (**D**) TNF-α in the back skin of mice from different groups. Scale bar: 100 μm. (**E**) Quantitative results of positive signals. The symbols *, &, and # indicate significant differences compared to the control group, BLE group, and BLE + VP1u group, respectively.

**Figure 7 ijms-24-15294-f007:**
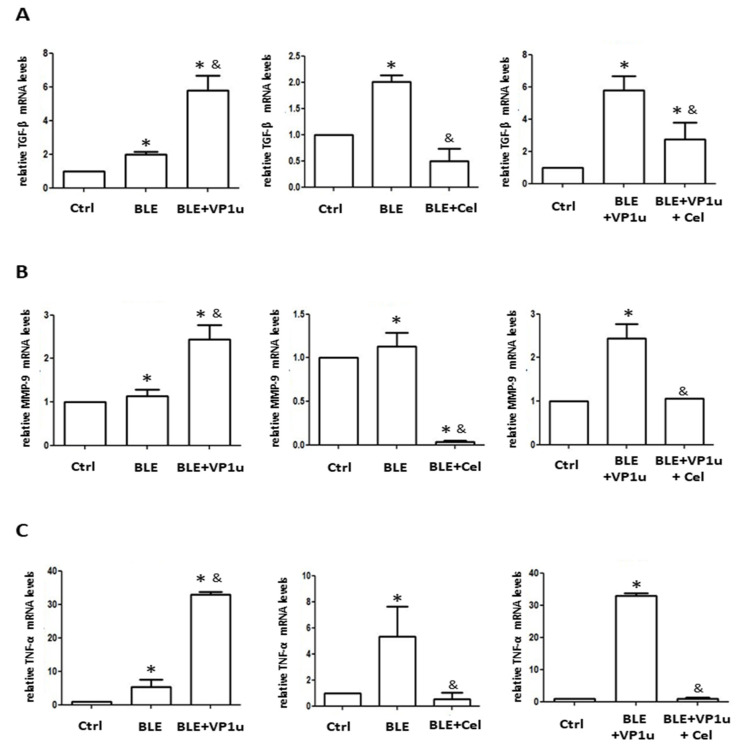
Expression of fibrosis-related genes in the skin tissues of mice. Relative mRNA expression levels of (**A**) TGF-β, (**B**) MMP-9, and (**C**) TNF-α in skin tissues of mice from different groups. The symbols * and & indicate significant differences compared to the control group, BLE group or BLE + VP1u group, respectively.

**Table 1 ijms-24-15294-t001:** Inhibition of secreted phospholipase A2 (sPLA2) activity of B19V-VP1u and bvPLA2 by celastrol.

Proteins	sPLA2 Activity (μmol/min/mL)
bvPLA2 (10 ng)	0.775 ± 0.020
bvPLA2 (10 ng) + Celastrol (0.5 μM)	0.328 ± 0.028 *
bvPLA2 (10 ng) + Celastrol (1.0 μM)	0.250 ± 0.045 *
bvPLA2 (10 ng) + Celastrol (2.0 μM)	0.201 ± 0.059 *
B19V-VP1u (1 μg)	0.306 ± 0.002
B19V-VP1u (1 μg) + Celastrol (0.5 μM)	0.182 ± 0.001 *
B19V-VP1u (1 μg) + Celastrol (1.0 μM)	0.090 ± 0.007 *
B19V-VP1u (1 μg) + Celastrol (2.0 μM)	0.062 ± 0.005 *
Celastrol (0.5 μM)	ND
Celastrol (1.0 μM)	ND
Celastrol (2.0 μM)	ND

bvPLA2: sPLA2 from bee venom PLA2 control; B19V: human parvovirus B19; VP1u: VP1 unique region; ND: non-detected. * Indicates a significance (* *p* < 0.05) as compared to control.

## Data Availability

The data presented in this study are available on request from the corresponding author.

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
