# Peer review of "Effect of the Functional VP1 Unique Region of Human Parvovirus B19 in Causing Skin Fibrosis of Systemic Sclerosis"

_ijms, 2023, doi:10.3390/ijms242015294_

Round 1
Reviewer 1 Report
Your original study reporting that the B19V-VP1u gene region is associated with skin fibrosis in systemic sclerosis was written in an understandable scientific language and was found suitable for publication in its current form.
Congratulations to you
My views on the article titled "Effect of the Functional VP1 Unique Region of Human Parvovirus B19 in Causing Skin Fibrosis of Systemic Sclerosis" are given below:
In the abstract, the relationship of B19V with systemic sclerosis is stated, and it is claimed that the exact mechanisms underlying the contribution of B19V to the development of systemic sclerosis remain unclear. It is understood that the study was planned to investigate this problem.
The summary of the study investigating the effects of B19V-VP1u on macrophages and bleomycin-induced SSc mice reflects the content of the article and is sufficient.
B19V and its genomic features are summarized in the Introduction; the functions of the B19V-VP1u gene region and its relationship with diseases are explained; and the main purpose of the study is stated based on sufficient literature knowledge.
In the Results section, the effects of phospholipase A2 activity of B19V-VP1u on human macrophages, the effects of B19V-VP1u on skin tissues in a mouse model of bleomycin-induced systemic sclerosis, the effects of B19V-VP1u-activated inflammatory responses in human macrophages, and the effects of B19V-VP1u on α-SMA, collagen I, and cytokine expressions in the skin tissues of a bleomycin-induced systemic sclerosis mouse model are described. 1 table and 7 figures in the results presented under four subheadings are directly related to the subject and necessary.
The discussion includes literature related to the subject. In their study, the authors noted that immunohistochemistry showed increased expression of fibrotic markers in dermal fibroblast cells following exposure to B19V-VP1u. Based on this information, they stated that the role of B19V, especially functional B19V-VP1u, in the pathogenesis of SSc provides rational evidence for its explanation.
Materials and Methods are presented under 14 subheadings, and it is understood that the original study was carried out with precise measurements.
In the results section, the authors first stated in their study that celastrol effectively reduced the Group XIII phospholipase A2 activity of B19V-VP1u and suppressed NF-κB and MMP-9 activity in B19V-VP1u-activated human macrophages. Furthermore, in an animal model of bleomycin-induced systemic sclerosis, B19V-VP1u application led to skin thickening accompanied by increased hydroxyproline levels. They reported that all of these were alleviated by celastrol treatment. Higher degrees of fibrosis are associated with higher levels of relevant fibrotic factors. In conclusion, this study reveals that B19V-VP1u exacerbates skin fibrosis in systemic sclerosis, a phenomenon that is effectively ameliorated by celastrol treatment.
15 of the 49 references used in the article are current literature from 2018–2022. Six articles (reference numbers 8, 11, 31, 33, 47, and 48) published in various journals by some of the authors (including the correspondent author) were used as references. This number is 12.2% of the references used. However, these self-references are directly relevant and necessary.
In conclusion, the study reporting that the B19V-VP1u gene region is associated with skin fibrosis in systemic sclerosis, is original and found suitable for publication in its current form.
Reviewer 2 Report
The Authors of the present study report that celastrol, a product derived from traditional Chinese medicinal plants with inflammatory, antioxidant, and anticancer properties, reduces sPLA2 activity and suppresses NF-κB and MMP-9 activity in parvovirus B19V-VP1u activated human macrophages of the bleomycin-induced systemic sclerosis animal model. Moreover, administration of parvovirus B19V-VP1u led to skin thickening accompanied by increased hydroxyproline expression that were reduced by celastrol treatment. Finally, B19V-VP1u increases the levels of fibrotic factors including TGF-β, α-SMA, collagen I, IL-6, IL-17A, IL-18, TNF-α, and MMP-9 . In summary, this study shows that parvovirus B19V-VP1u exacerbates skin fibrosis in SSc that may be reduced by celastrol treatment. Although of potential interest, these data refer to an animal model, therefore the Authors should better discuss their potential transferability to human systemic sclerosis.
The English Language only needs minor editing
Round 2
Reviewer 2 Report
The Authors addressed in the revised version the queries of the Reviewer.